# TRAINING A CONSTRAINED NATURAL MEDIA PAINTING AGENT USING REINFORCEMENT LEARNING

## ABSTRACT

We present a novel approach to train a natural media painting using reinforcement learning. Given a reference image, our formulation is based on stroke-based rendering that imitates human drawing and can be learned from scratch without supervision. Our painting agent computes a sequence of actions that represent the primitive painting strokes. In order to ensure that the generated policy is predictable and controllable, we use a constrained learning method and train the painting agent using the environment model and follows the commands encoded in an observation. We have applied our approach on many benchmarks and our results demonstrate that our constrained agent can handle different painting media and different constraints in the action space to collaborate with humans or other agents.

## 1 INTRODUCTION

Throughout human history, painting has been an essential element of artistic creation. There are many diverse and complex artistic domains with various styles such as watercolor, oil painting, sketching, and so on. As image processing and computer graphics have advanced, there has been a considerable effort to simulate these styles using non-photorealistic rendering (NPR) techniques (Kumar et al. (2019)). Hertzmann (1998); Winkenbach & Salesin (1996) generate compelling results using stroke-based rendering. However, most prior methods in NPR are engineered for a specific application or task, and cannot easily adapt to new styles or medium.

Recent developments in machine learning have resulted in significant advancements in computer vision and computer graphics, including computer-based painting systems. Many visual generative methods based on generative adversarial networks (Goodfellow et al. (2014)) as Zhu et al. (2017); Zhou et al. (2018); Huang et al. (2018); Karras et al. (2017); Sangkloy et al. (2017) have demonstrated promising results. Many of these machine learning methods have also been applied to stroke-based rendering tasks, including modeling the brush (Xie et al. (2012); Zheng et al. (2019)), generating brushstroke paintings in an artist's style (Xie et al. (2015)), reconstructing drawings for specific paintings styles (Tang et al. (2018)), and constructing stroke-based drawings (Ha & Eck (2017a); Zhou et al. (2018); Huang et al. (2019); Jia et al. (2019a)).

In this paper, we focus on a more general and challenging problem of training a natural media painting agent for interactive applications. Given a reference image, our goal is to develop a stroke-based rendering approach that can imitate the human drawing or strokes used in generating the image. A key challenge is to develop a method that can learn from scratch without any supervision. In this regard, we present a technique that can handle all inputs and train an agent to manipulate natural painting media such as charcoal, pencil, watercolor, and so on. We build a model-based natural media environment using deep CNN and train a natural media painting agent using model-based reinforcement learning. In order to introduce controls to the agents for interactive applications, we use a constraint representation along with a different framework for training and use the constrained painting agent. These constraints enable the agent to interact with a human or other agents and generate various styles without retraining the model. The *novel contributions* of our work include:

- A method to train an agent that produces a stream of actions subject to constraint for each action. These constraints can include restricting the start location, stroke width, color, and other stroke parameters.

- A method to roll out constrained agents so the user can produce new stylistic effects interactively or automatically, as the agent is painting by modulating the action stream.
- By incorporate coarse-to-fine strategy, our painting agents can generate high-resolution stylized images using various constraints and paintbrush configurations.

We evaluate our algorithm on different paintbrush configurations and datasets to highlights its benefits over prior reinforcement learning based methods. We also employ differing constraint settings to validate our constrained agents and produce new stylistic effects with a single trained model.

## 2 RELATED WORK

In this paper, we focus on the stroke-based rendering problem, which renders the reference image with brush strokes. In contrast to image analogy approaches (Hertzmann et al. (2001); Gatys et al. (2015); Vondrick et al. (2016); Zhu et al. (2017); Karras et al. (2017); Sangkloy et al. (2017); Li et al. (2017; 2018)), stroke-based approaches can generate intermediate painting states for interactive purposes. They can also be deployed in various painting environment to generate different artistic effects.

One category of approaches for stroke-based rendering uses a heuristics-based method to guide the agent while painting. Notable examples include the method in Winkenbach & Salesin (1996), which can produce stylized illustrations, and the method in Hertzmann (1998), which can reproduce a colorful painting in different styles. However, it is difficult to extend these methods to different styles of painting because of the hand-engineered features.

Another category of approaches uses machine learning techniques to learn the policy. Compared with predefined policies, these methods use machine learning techniques to learn the painting policy, which enables the agent to generate a more natural result. Ha & Eck (2017b) train an RNN to learn the latent space of the sketch data and generate the paintings in this latent space, which requires the paired dataset for training.

Other approaches use deep reinforcement learning to learn the policy without supervised data. There have been a few attempts to tackle related problems in this domain. Xie et al. (2012; 2015; 2013) propose a series of works to simulate strokes using reinforcement learning and inverse reinforcement learning. These prior approaches learn a policy either from reward functions or expert demonstrations. Comparing with these approaches, we use a more general setup which does not rely on rewards engineering.

Zhou et al. (2018) is based on the Deep Q network. Ganin et al. (2018) trains the discriminator and the reinforcement learning framework at the same time. However, both methods can work on either a small action space or a small observation space. Jia et al. (2019b;a) use proximal policy optimization with curriculum learning and self-supervision to gradually increase the sampled action space and the frequency of positive rewards. Zheng et al. (2019) train a differentiable painting environment model, which helps learn the painting policy with gradient-based methods. Huang et al. (2019) train a model-based RL using differentiable environment and DDPG (Lillicrap et al. (2015)). This approach is limited by the OpenCV based renderer and the uncontrollable agent.

## 3 OVERVIEW

An overview of our approach is given in Fig. 1 This includes the training phase and extracting the underlying neural network as the painting agent is used for the roll-out process. We highlight all the symbols used in the paper in Table 1.

In section 4, we present the details of natural media painting agent with a environment model for the agent to interact. Our approach is based on the training scheme described in Huang et al. (2019) and uses Deep Deterministic Policy Gradient (DDPG) (Lillicrap et al. (2015)) to train the model-based painting agent. For the roll-out algorithm, we apply the painting agent directly to the real environment $R_r$ as shown in Figure 1. In section 5, we present our representation of the underlying constraints and the underlying techniques to encode and decode these constraints using reinforcement learning. We use the unconstrained agent to identify the constraints, and encode them as observations to train the constrained painting agent, as shown in Figure 3. In order roll out the

Table 1: Notation Summary

| Symbol | Meaning | Symbol | Meaning |
|---|---|---|---|
| $t$ | step index | $s_t$ | current painting state of step $t$, canvas |
| $s^*$ | target painting state, reference image | $\hat{s}^*$ | reproduction of $s^*$ |
| $c_t$ | constraint of step $t$ | $o_t$ | observation of step $t$ |
| $a_t$ | action of step $t$, $a_t = [\alpha_t, l_t, w_t, c_t]$ | $r_t$ | reward of step $t$ |
| $q_t$ | accumulated reward of step $t$ | $\gamma$ | discount factor for computing the reward |
| $\pi$ | painting policy, predict $a$ by $o$ predict $r$ by $o$ | $V_\pi$ | value function of the painting policy, |
| $R(a_t, s_t)$ | render function, render action to $s_t$ state and the target state | $O(s^*, s_t)$ | observation function, encode the current |
| $R_r$ | real renderer | $R_n$ | neural renderer |
| $L(s, s^*)$ | loss function, measuring distance between state $s$ and objective state $s^*$ | | |

constrained agent, we replace the unconstrained agent with user defined constraints or the action of previous step.

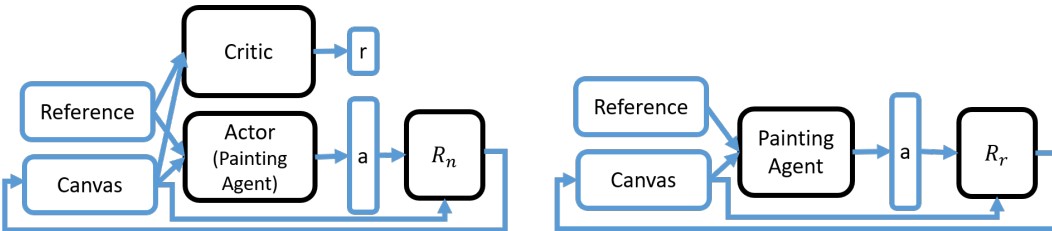

Figure 1: *Training the Natural Media Painting Agent (left):* We use a actor-critic-based reinforcement learning framework to train the painting agent. For each step, the current states of the canvas and the reference image form the observation for the policy network. Based on the observation, the painting agent predicts an action for the neural renderer $R_n$ to execute and updates the canvas accordingly. *Roll-out of the Natural Media Painting Agent (right):* We extract the actor neural network as the painting agent for the roll-out process. We replace the neural renderer $R_n$ with the real renderer $R_r$ to get the precise synthetic results.

## 4 NATURAL MEDIA PAINTING AGENT

In this section, we present our algorithm for training a natural media environment model and training a painting agent to interact with this environment model.

### 4.1 RENDERER

The renderer of our painting agent generates the corresponding canvas by the given action, which implements the blending functions and other synthetic programs. Unlike the previous learning-based painting approaches Jia et al. (2019a); Huang et al. (2019); Zheng et al. (2019), we use a natural media painting renderer MyPaint (libmypaint contributors (2018)) for our experiment. Compared with self-defined painting simulators, it provides rich visual effects and sophisticated blending functions. Ganin et al. (2018) use the same environment but have not explored the various configurations of the paintbrush. Using a pre-existing environment setup helps our approach concentrate on the learning algorithm to make it easy to generalize.

### 4.2 ACTION REPRESENTATION

The action of our painting agent consists of the configurations it use to interact with the environment. For the action space, we follow the stroke definition of Ganin et al. (2018) using a quadratic Bezier curve (QBC) of three points $((x_0, y_0), (x_1, y_1), (x_2, y_2))$. Each stroke is a 3-point curve. The pressure which affects the blending function is linear interpolated by the value of the start position $(x_0, y_0)$ and the end position $(x_2, y_2)$. We use one color $(R, G, B)$ for each stroke, and the transparency of the pixels within the stroke is affected by the pressure $(p_0, p_1)$. We use one brush size

$r$ for each stroke, and the actual stroke width within the stroke can also be affected by the pressure $(p_0, p_1)$. Formally, we represent the action $a_t$ as:

$$a_t = (x_0, y_0, x_1, y_1, x_2, y_2, p_0, p_1, r, R, G, B) \tag{1}$$

In practice, $a_t \in \mathbb{R}^{12}$. Each value is normalized to $[0, 1]$. The action is in a continuous space, which makes it possible to control the agent precisely.

### 4.3 OBSERVATION REPRESENTATION

The observation of our painting agent consists of information to make decisions. In our setup, the observation is represented by the reference image $s^*$, the current canvas $s_t$ at step $t$, and the constraint $c_t$. Formally, $o_t = (s^*, s_t)$. For the constrained agent described in section 5, $o_t = (s^*, s_t, c_t)$. $c_t$ is a vector drawn from the sub-space of the action space. To unify the observation representation for implementation, we upsample the constraint vector as a bitmap with size of $s_t$ to feed into the CNN of the policy network and the value network.

### 4.4 REWARD

Reward is a metric that enables our painting agent to measure effectiveness of the action. Aside from the reinforcement learning algorithm, the definition of the reward can be seen as a guidance for the policy $\pi$. For our problem, the goal of the painting agent is to reproduce the reference image $s^*$. First, we define the loss function $L$ between the current canvas $s_t$ at step $t$ and the reference image $s^*$. In practice, we use $l_2$ loss (Jia et al. (2019a)) and WGAN loss (Huang et al. (2019)) for implementation. Then, we use the difference of the losses of two continuous steps as the reward $r_t$. We normalize $r_t$ using Eq.2, such that $r_t \in (-\infty, 1]$.

$$r_t = \frac{L(s_{t-1}, s^*) - L(s_t, s^*)}{L(s_0, s^*)}, \tag{2}$$

where $L$ is a loss function defined as $l_2$ or WGAN. For our reinforcement learning setting, the objective is to maximize the discounted accumulative rewards $q_t = \sum_{t=1}^{t_{max}} r_t \gamma^t$, where the discount factor $\gamma \in [0, 1]$

### 4.5 ENVIRONMENT MODEL

For this painting environment, we use the rendering function $R$ to represent the transition function. The action only modifies the current painting state $s_t$ with the current action $a_t$ as $s_{t+1} = R(a_t, s_t)$. Inspired by Huang et al. (2019) and Zheng et al. (2019), we model the behaviors of the real environment renderer $R_r$ using a neural network as $R_n$. We use a 4-layer fully connected neural network followed by 3 convolutional neural networks.

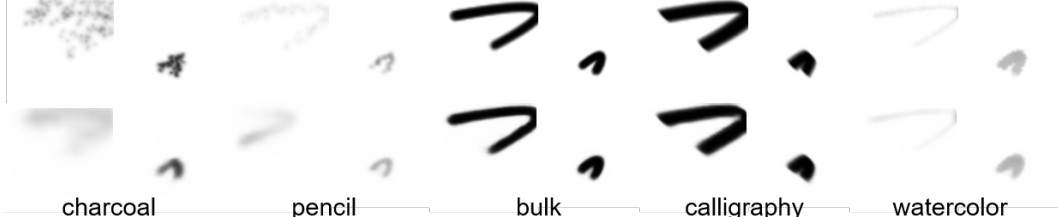

charcoal  pencil  bulk  calligraphy  watercolor

Figure 2: *Environment Model* We model the painting environment MyPaint(libmypaint contributors (2018)) with different paintbrush configurations. The top row is rendered by the real renderer $R_r$ and the bottom row is rendered by the neural renderer $R_n$.

There are two main benefits of modeling the environment. First, the reinforcement learning algorithm may have millions of steps, and it can be computationally expensive to use the real environment because of sophisticated synthetic algorithms. If we simplify the environment model, it can save training time. Second, modeling the transition function of states using neural network makes the entire framework differentiable, and the loss can be backpropagated through the renderer network $R_n$.

In practice, we model $R_r$ by building a dataset consisting of paired data $(a_t, s_t, s_{t+1})$. To simplify the problem, we always start from a blank canvas $s_0$ so that the $R_n$ only need to predict the stroke image from a single action. At run-time, we synthesize the blending function by making the stroke image transparent and adding it to the current canvas $s_t$. The paired dataset $\{a_{(i)}, s_{(i)}\}$ can be collected by randomly sampling the $a_i$ and synthesising the corresponding $s_i$.

We treat different paintbrushes of MyPaint (libmypaint contributors (2018)) as separate models. It is worth noting that we use the environment model $R_n$ for the training process, but use the real environment $R_r$ for the roll-out process.

## 5 Constrained Painting Agent

After we train the natural media painting agent, we make the agent controllable to handle interactive scenarios and be used for generating stylized images.

To achieve this goal, one straightforward way is to constrain the agent with a modified action space. However, this method is difficult to generalize to different constraints. For each different constraint $c$ in the same subspace $C$ in action space $A$, the policy needs to be retrained. For example, if we train a painting agent and constrain its stroke to width $0.8$, we still need to train it again when we need an agent to output a stroke with width $0.9$.

In the rest of this section, we propose a constrained painting agent that, can follow any constraint $c$ drawn from the subspace of action space $A$. First, we define the constraint representation, including the definition of the constraint, the encoding function, and the decoding function. Then we introduce the training and roll-out schemes of the constrained agent.

### 5.1 Constraint Representation

The constraint is the vector $c$ in the constraint space $C$, which is the sub-space of the action space $A$. While fixing the vector $c$, the agent has to explore the subtraction of $A$ and $C$ as $A'$ by sampling $a'$ from $A'$ and concatenating with $c$:

$$a = a' \oplus c, a' \in A' = A - C, c \in C \tag{3}$$

We can define $C$ by selecting and combining dimensions in $A$. For example, we can constrain the color of the stroke by defining $C$ as the color space $(R, G, B)$. We can constrain the start position of the stroke by defining $C$ as $(x_0, x_1)$. We can constrain the stroke width of the stroke by defining $C$ as $(r)$. Moreover, we can use the combination of the constraints to achieve complex effects.

As shown in Figure 3, the unconstrained agent takes the reference image and the current canvas $(s^*, s_t)$ as observation while the constrained agent takes the reference $(s^*, s_t, c_t)$ as an observation, which has an additional constraint $c_t$ at step $t$. $c_t$ is a vector drawn from the sub-space of the action space. To unify the observation representation for implementation, we encode the constraint vector $c_t$ as a bitmap to feed into the CNN of the policy network and the value network.

To encode the constraint $c_t$ into observation, we upsample the constraint $c_t$ as a matrix $c'_t$ which has the same size of $s^*$ and $s_t$, to stack them and feed into our policy network and our value network. To decode the constraint $c'_t$ into $c_t$, we implement a downsample module within the policy network. The policy network $pi$ can be seen as separate policy: $\pi_c$ and $\pi_a$; $\pi_c$ outputs the downsampled constraint $c$ and $\pi_a$ outputs the constrained action $a'$. Then we concatenate $a'$ and $c$ to form action $a$.

Formally, we have $c_t = \pi_c(c'_t)$, $a'_t = \pi_a(s^*, s_t, c'_t)$, and $a_t = a'_t \oplus c_t$.

### 5.2 Training the Constrained Painting Agent

After we introduce the constraint $c_t$ as part of the observation of the painting agent, we propose the corresponding training scheme. Because the constraint representation is designed for interactive purposes, we need to use either human experts or another agent to train the constrained agent.

As shown in Figure 3, we use an unconstrained agent to generate constraints by cascading them together. For each step $t$, the unconstrained agent takes the reference image and the current canvas

$(s^*, s_t)$ as observations, and outputs the action $a$ in action space $A$. Then we identify a subspace in $A$ as constraint space $C$ and transfer $a_t$ to $c_t$, followed by upsampling the $c_t$ as $c'_t$, as defined in section 4. After that, the constrained agent takes the additional constraint $c'_t$ as an observation $(s^*, s_t, c'_t)$ and outputs action $a_t$ concatenated by $c_t$ and $a'$.

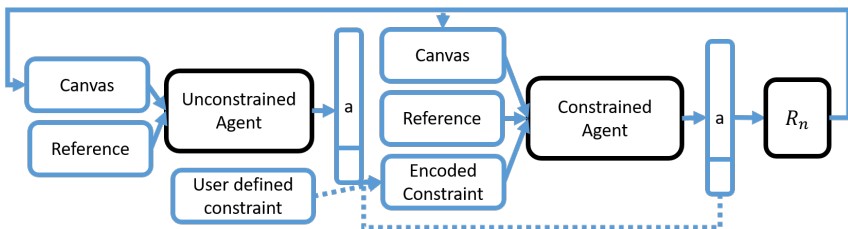

Figure 3: *Framework for Training (Roll-out) the Constrained Agent* We train the constrained agent by cascading an unconstrained agent. First, we identify a subspace of the action space and extract the constraint $c$ from the action $a$. Then we upsample the constraint $c$ and pass it on as an observation of the constrained agent. We use $R_n$ as renderer. For roll-out process (dashed line), we extract the constraint from either the user defined constraint or the action of the previous step. We use real renderer $R_r$ to get the best visual quality.

## 5.3 ROLL-OUT CONSTRAINED PAINTING AGENT

For the roll-out of the constrained painting agent, we replace the unconstrained agent from the training scheme in Figure 3 with the constraint extracted from either human command or a painting agent's output for each step $t$.

When the constraint is extracted from a human command, the agent can be ordered to place a paint stroke with a given position $(x_0, y_0)$, color $(R, G, B)$, and stroke width $r$. When the constraint is extracted from painting agent's output, it can be seen that we split the one agent that explores in entire action space $A$ into two agents that explore separate action spaces $A_0$ and $A_1$, where $A = A_0 + A_1$.

## 6 RESULT

### 6.1 NATURAL MEDIA PAINTING ENVIRONMENT MODEL

For training the natural media painting environment, we treat different paintbrushes as different models. We run 50,000 iterations for each model and record the $l_2$ loss on the validation dataset as shown in Table 2. The learning curves of the training processes are shown in Figure 4 (left).

Table 2: $l_2$ *loss of paintbrush models*

| Paintbrush | $l_2$ **Loss** | Paintbrush | $l_2$ **Loss** |
|---|---|---|---|
| charcoal | $2.12 \times 10^{-3}$ | impressionism | $2.43 \times 10^{-3}$ |
| pencil | $8.37 \times 10^{-5}$ | marker | $5.01 \times 10^{-4}$ |
| bulk | $9.16 \times 10^{-4}$ | dry brush | $3.06 \times 10^{-3}$ |
| calligraphy | $5.92 \times 10^{-4}$ | watercolor | $1.16 \times 10^{-4}$ |

### 6.2 UNCONSTRAINED PAINTING AGENT

We train the unconstrained painting agent using the fixed enviroment model with various dataset. We use hand-written digits images (MNIST LeCun & Cortes (2010)), character images (KanjiVG Apel (2014), face images (CelebA Liu et al. (2015), object images (ImageNet Deng et al. (2009)) as train painting agents. We run 10,000 episodes for each training task. Because of the difference among the dataset, we use 5 strokes to reproduce hand-written digits images, 20 strokes to reproduce character images, 100 strokes to reproduce face and object images.

We measure the $l_2$ loss between the reference images and reproduced images throughout the training process shown as Figure 4(middle). The reproduced results is shown as Figure 5, and the corresponding $l_2$ loss is shown as Table 3(left).

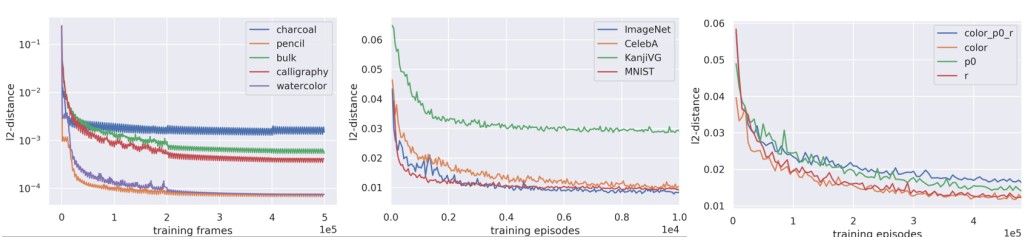

Figure 4: $l_2$ *Loss over the Course of Training.* We compute the $l_2$ distance between the roll-out results and the reference images on a validation dataset. (Left) is the course of training the neural renderer $R_r$ using various paintbrush. (Middle) is the course of training the unconstrained agents using varisous dataset. (Right) is the course of training the constrained agents using various constraints configurations.

Table 3: $l_2$ *training loss of training schemes*

| Training Scheme: Dataset | $l_2$ Loss | Training Scheme: Constraint | $l_2$ Loss |
|---|---|---|---|
| MNIST | $8.16 \times 10^{-3}$ | width $(r)$ | $1.25 \times 10^{-2}$ |
| KanjiVG | $4.02 \times 10^{-2}$ | start position $(x_0, y_0)$ | $1.43 \times 10^{-2}$ |
| ImageNet | $7.06 \times 10^{-3}$ | color $(R, G, B)$ | $1.18 \times 10^{-2}$ |
| CelebA | $8.55 \times 10^{-3}$ | all above $(x_0, y_0, r, R, G, B)$ | $1.63 \times 10^{-2}$ |

|  |  |  |  |
|---|---|---|---|
| MNIST | KanjiVG | CelebA | ImageNet |

Figure 5: We trained our natural media painting agents using MNIST, KanjiVG, CelebA, and ImageNet as reference images (left). We generate results (right) using $R_r$ for the training and validating process.

For the roll-out process, we employ a coarse-to-fine strategy to increase the resolution of the result shown as Figure 6. First, we roll out the agent with a low-resolution reference image $s^*$ and get the reproduction $\hat{s^*}$. Then we divide $s^*$ and $\hat{s^*}$ into patches and feed the agent as initial observation. The roll-out results using various natural media are shown as Figure 7.

## 6.3 CONSTRAINED PAINTING AGENT

We train the constrained painting agents using the learning parameters as unconstrained painting agents. We compute the $l_2$ distance between the reference images and reproduce results for the training process with various constraint configurations. To control variates, we use same neural environment $R_n$ (charcoal) and dataset (CelebA) for these experiments. We use the color $(R, G, B)$, the stroke width $r$, the start position $(x_0, y_0)$ and all of them $(x_0, y_0, r, R, G, B)$ as constraints.

The learning curves of these constraints are shown as Figure 4(right) and the $l2$ loss is shown as Table 3(right). We demonstrate the constrained painting agents as Figure 2, which uses pencil as paintbrush and incorporates coarse-to-fine strategy by dividing the reference images as $4 \times 4$ patches. To constrain the color of strokes, we build a color palette by clustering the colors of the reference image. For each action $a$, we constrain it from a color randomly selected from the color palette. To constrain the stroke width, we use a constant stroke width for the roll-out process.

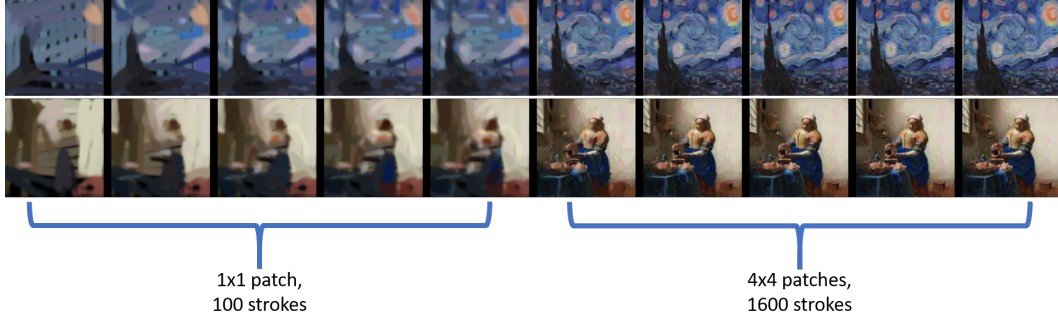

1x1 patch,
100 strokes

4x4 patches,
1600 strokes

Figure 6: *Coarse-to-fine Roll-out* We roll out the trained agent using a coarse-to-fine strategy. We first roll out the model treating the reference image and canvas as one patch. Then we divide the reference image and updated canvas into patches and feed to the agent. In this figure, we roll out 100 strokes in each patch.

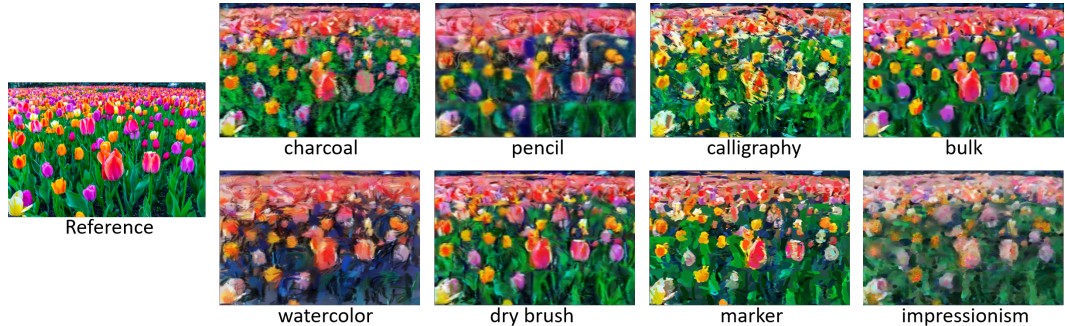

Figure 7: *Exemplary Result* Results generated by the painting agent using different paintbrushes. We demonstrate our natural media painting agent by rolling out agents trained with various paintbrushes from MyPaint.

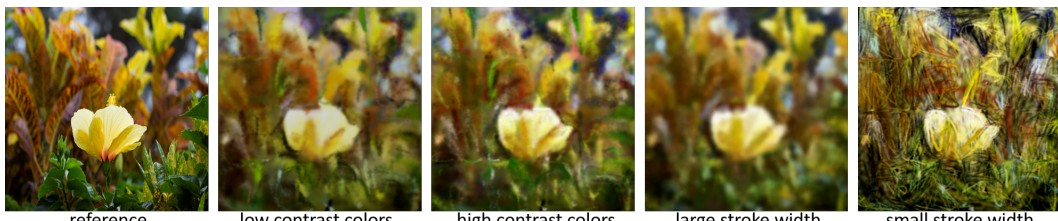

Figure 8: *Roll-out Using Constrained Painting Agents* We constrain the stroke width and the color of the agent to get the stylized results.

## 7 CONCLUSION AND FUTURE WORK

In this paper, we train natural media painting agents that can generate artistic paintings using various natural media, and collaborate with humans and other agents to get different visual effects. We build a model of natural media environment using deep CNN and train a natural media painting agent using model-based reinforcement learning. To introduce controls to the agents for interactive purposes, we propose constraint representation, a framework for training a constrained painting agent, and various roll-out schemes to apply the agent. We demonstrate our algorithm by applying the trained model using various paintbrushes from MyPaint and constraints set up. The experimental results show that our algorithm can reproduce reference images in multiple artistic styles.

For future work, we aim to extend the proposed algorithm by building a unified model for differing paintbrush configuration. In addition, we will train a hierarchical agent that uses a constrained agent as the low-level policy. We would like to apply our approach on other reference images and use for interactive painting systems.

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

## A  APPENDIX

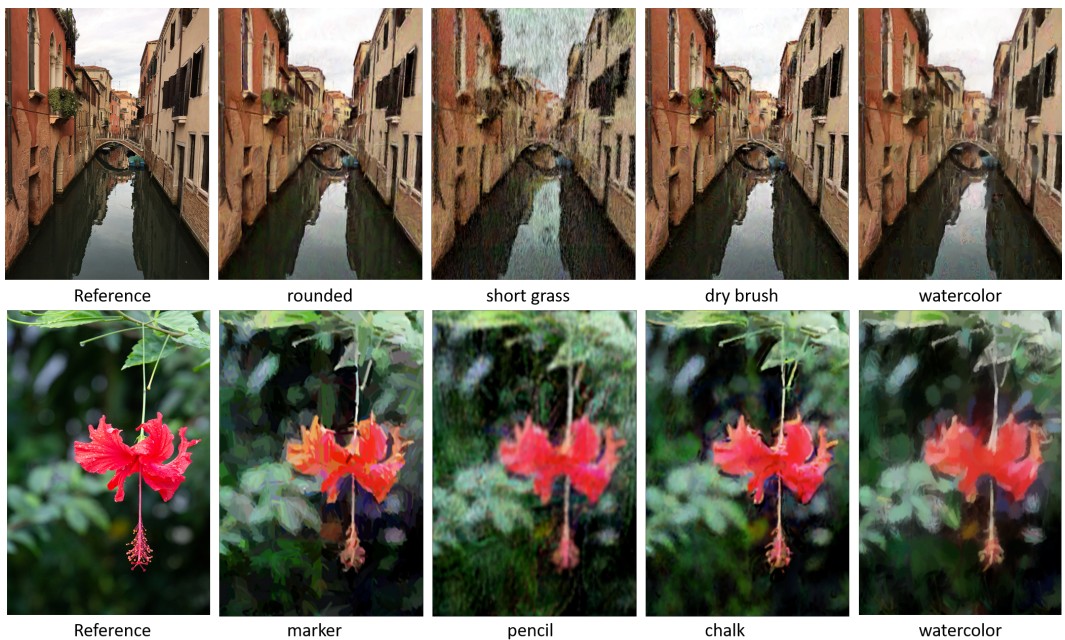

Figure 9: Roll-out results using Various PaintbrushesWe roll out our natural media painting agents trained with various brushes in MyPaint. To increase the resolutions of the generated images, we incorporate the coarse-to-fine strategy. We use $8 \times 8$ patches for first row and $4 \times 4$ for second row.

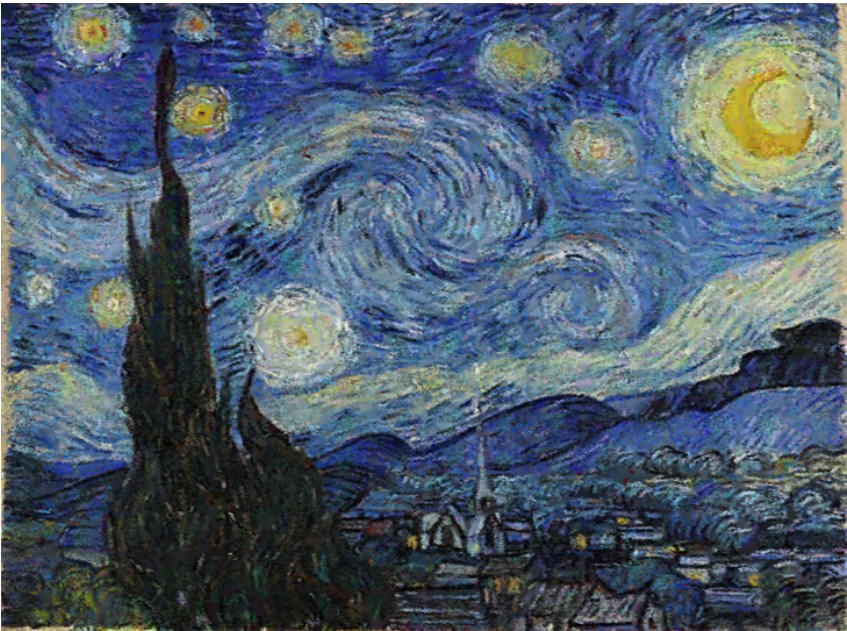

Figure 10: *Reproduction of Starry Night using Charcoal* We roll out our natural media painting agent trained with charcoal brush in MyPaint to reproduce Van Gogh's starry night.We incorporate the coarse-to-fine strategy by dividing the reference image and canvas into $16 \times 16$ patches.

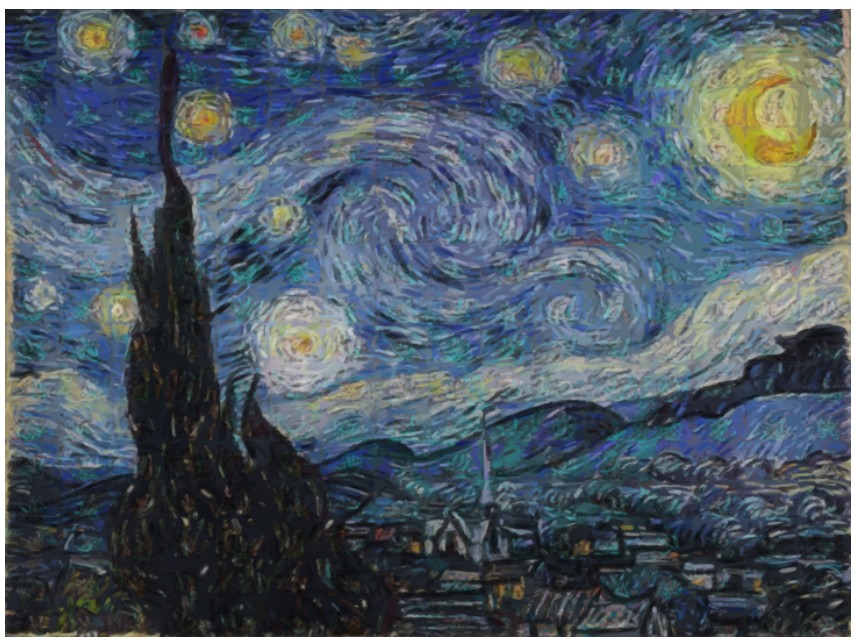

Figure 11: *Reproduction of Starry Night using Watercolor* We roll out our natural media painting agent trained with watercolor brush in MyPaint to reproduce Van Gogh's starry night.We incorporate the coarse-to-fine strategy by dividing the reference image and canvas into $16 \times 16$ patches.

