# OpenReview forum: "Training a Constrained Natural Media Painting Agent using Reinforcement Learning "
_ICLR.cc/2020/Conference — Reject_

### Official Review · AnonReviewer3 · 2019-10-22
**Official Blind Review #3**

**Rating:** 3

**Review:**

The authors present the results of training a natural media painting agent using reinforcement learning for different types of strokes. The agent seems to be capable of learning how to pain under different types of constraints and produce visually interesting images.

Comments:

- Given that the authors give an implementation of the constrained RL agent as one of the key contributions of the paper, there is a glaring absence of mentioning related work on constrained reinforcement learning and reviewing the existing approaches in literature, in order to compare and contrast what the authors propose in this paper. This makes it hard for the readers to assess the novelty of the contribution.

- Similarly, the authors should discuss in more detail the limitations on the types of constraints that are possible to easily express in this framework, given the simplicity of the constraints that are shown in the experiments

- The readability of the paper seems like it could be improved. Apart from typos, there seem to be many long enumerations of approaches that other researchers have taken in this space, but for a reader it is not immediately obvious how these come together and relate to the work that is being presented.

- With the above in mind, one thing that was conceptually unclear to me is that one of the main advantages of the proposed approach, according to the authors, when compared to some of the cited related work, is that this approach can generate intermediate representations and not just the final output with most resemblance to the reference picture. There is a mention of this opening up new artistic possibilities. Yet, this particular use case is not given central stage in evaluation. The authors should provide more examples of why this particular capability is relevant and how it leads to interesting outcomes.

- The authors say: “we use 5 strokes to reproduce hand-written digits images, 20 strokes to reproduce character images, 100 strokes to reproduce face and object images” - which does seem reasonable, but the actual numbers (5, 20, 100) are not well motivated. After all, why not (10, 50, 200) or (5, 40, 100) or (5, 40, 200)? It would be good to show experimentally (for one of these) that the choice is justified. I’m not familiar with KanjiVG - but there do exist kanji characters with more than 50 strokes. I’m guessing they are not present in this dataset?

- The authors should provide more details on the specifics of the model architecture and the hyperparameters that have been used / explored.

- In the results/discussion of the paper, the authors should compare to prior work and highlight their novel contributions. Given that multiple papers out there that have had at first glance similar-looking results, it is hard to otherwise qualitatively judge whether what is proposed in this paper is better, without any form of side-by-side comparison.


**Experience Assessment:**

I have read many papers in this area.

**Review Assessment: Checking Correctness Of Derivations And Theory:**

I assessed the sensibility of the derivations and theory.

**Review Assessment: Checking Correctness Of Experiments:**

I carefully checked the experiments.

**Review Assessment: Thoroughness In Paper Reading:**

I read the paper at least twice and used my best judgement in assessing the paper.

---

### Official Review · AnonReviewer2 · 2019-10-23
**Official Blind Review #2**

**Rating:** 1

**Review:**

This paper presents a reinforcement learning agent trained to interact with a painting environment in order to reproduce target images. The novel aspect of this work seems to be a version of the agent producing partial actions (i.e., portion of the action tuple is clamped to pre-specified values). This new agent receives the clamped components of the action as an additional conditioning input.

Pros:
+ In some cases, images produced by the system look appealing.

Cons:
- The writing of the manuscript could be significantly improved. I had a hard time understanding certain parts of the paper (e.g., what “constrained” means in the context of the present work). I got an impression that there was an effort to make things look more complex than they really are.
- The proposed model lacks novelty – there seems to be only one non-trivial contribution and I’m not entirely sure how useful it is. The authors never compare their system against a simple baseline when one just overrides the actions of the agent.
- In general, the evaluation section of the paper leaves a lot to be desired. The authors report some numbers (most of which are not even for the proposed model) but I’m struggling to make anything out of that information. Why should I care about them? What interesting conclusions can I draw from them? The paper never discusses this. On top of that, there are no baselines.

Notes/questions:
* At test time, the policy receives renders from a real environment. Could that create problems since it has only been trained on images synthesized by a neural surrogate?
* Abstract: “on many benchmarks” -> “on several benchmarks”
* Section 1: “we use a constraint representation along with …” – this sentence needs to be rewritten.
* Section 2: Missing reference – Neural Painters (Nakano, 19). Considers a neural surrogate of the libmypaint environment.
* Section 2, last paragraph:  “both methods can work on either a small action space or a small observation space” – how is the present model different? The action space is very similar (albeit continuous) to the existing approaches (i.e., relies on Bezier curves much like in (Ganin et al., 18)).
* Section 2, last paragraph: The last sentence needs to be rewritten (“uncontrollable agent” looks a bit strange)
* Section 3, first paragraph: “We highlight all …” -> “We describe all …”
* Section 4.1, first paragraph: The first sentence needs to be rewritten (“the corresponding canvas by the given action”).
* Section 4.1, first paragraph: “Unlike the previous …” – not true. (Ganin et al., 18) proposes to use this environment and (Nakano, 19) trains a neural surrogate for it.
* Section 4.4: “WGAN loss (Huang et al., 19)” – this WGAN loss was introduced in (Ganin et al., 18).
* Section 5: I feel like the paper could do a better job at explaining what “constraining” really means and justifying why it’s an interesting problem to solve. In my opinion, Eq. (3), for example, obscures rather than clarifies the notion of “constraints”.
* Section 5, paragraph 2: “For each different” -> “For each”
* Section 5.1, paragraph 4: “pi” -> “\pi”
* Figure 5: The caption almost overlaps with the text below.
* Section 6.3, paragraph 2: “l2” -> “l_2”
* Section 7, paragraph 2: “differing” -> “different”

I feel like the authors should perform a major re-writing of the manuscript before it’s ready for publication. Moreover, I failed to see any significantly novel aspects of the proposed system (maybe due to the poor presentation) and therefore I wouldn’t recommend the paper for acceptance.

**Experience Assessment:**

I have published one or two papers in this area.

**Review Assessment: Checking Correctness Of Derivations And Theory:**

I assessed the sensibility of the derivations and theory.

**Review Assessment: Checking Correctness Of Experiments:**

I assessed the sensibility of the experiments.

**Review Assessment: Thoroughness In Paper Reading:**

I read the paper at least twice and used my best judgement in assessing the paper.

---

### Official Review · AnonReviewer4 · 2019-10-30
**Official Blind Review #4**

**Rating:** 1

**Review:**

This paper proposes an RL agent for generating a painting from a photograph by optimizing a sequence of brush strokes to match the target image.

I believe the paper should be rejected because it does not have significant technical novelty for a first-tier conference, and the results do not show much aesthetic or technical advance. In terms of technical novelty, the paper seems to be applying a standard RL agent to an existing problem space, to optimize existing losses. The addition of constraints is technically very simple. The paper itself fails to articulate a compelling statement of novelty, for example, in discussing Huang 2019, the paper just says that a limitation of that method is that it uses OpenCV and that it doesn’t provide a system for control. Removing OpenCV is not a publishable contribution and shouldn’t even be mentioned; the control mechanism is not particularly novel.

Some of the results do look nice, but it’s hard to say that the method has improved over the past 20 years of stroke-based rendering, e.g., many of the results look worse than those in Hertzmann 1998. The paper doesn’t offer any meaningful comparisons to the previous work, such as fair side-by-side comparisons on the same images, comparing computation times and aesthetics. The paper doesn’t articulate any aesthetic goals or state any meaningful standard by which the images might have improved over previous work. The control mechanism is not tested in any meaningful way (e.g., user studies), and the changes it makes to the results seem fairly trivial (e.g., contrast can be added to an image just as well via a post-process).

I’m not sure what ICLR’s policy on citing previous unrefereed work is, but one paper that seems to have much more interesting results in a similar context is: https://arxiv.org/abs/1904.08410 . Reiichiro Nakano, Neural Painters: A learned differentiable constraint for generating brushstroke paintings.

**Experience Assessment:**

I have published in this field for several years.

**Review Assessment: Checking Correctness Of Derivations And Theory:**

I did not assess the derivations or theory.

**Review Assessment: Checking Correctness Of Experiments:**

I assessed the sensibility of the experiments.

**Review Assessment: Thoroughness In Paper Reading:**

I made a quick assessment of this paper.

---

### Note · Authors · 2019-12-02
**Submission Withdrawn by the Authors**

I have read and agree with the venue's withdrawal policy on behalf of myself and my co-authors.

---

### Decision · Program_Chairs · 2019-12-19

**Decision:**

Reject

**Comment:**

Paper is withdrawn by authors.